# Winter Wheat Seeding Decisions for Improved Grain Yield and Yield Components

Lawrence Aula [1], Amanda C. Easterly [2] and Cody F. Creech [1,*]

1    Panhandle Research, Education and Extension Center, Department of Agronomy and Horticulture, University of Nebraska-Lincoln, Scottsbluff, NE 69361, USA
2    High Plains Ag Lab, Department of Agronomy and Horticulture, University of Nebraska-Lincoln, Sidney, NE 69162, USA
*    Correspondence: ccreech2@unl.edu

**Abstract:** The continual re-evaluation of agronomic practices is necessary to improve crop performance and sustainability of the production of winter wheat (*Triticum aestivum* L.), particularly as genetics and climate conditions change. Recommendations made about winter wheat planting dates, spacing, variety, and seed rates under normal climatic conditions may not be suitable in current times with more climate variability. Our experiment investigated the effect of planting date (early, historic-optimum, and late), row spacing (19 and 25 cm), variety (Goodstreak, Robidoux, and Wesley), and seed rate (1.8, 2.1, 2.3, 2.4, 2.6, 2.8, 3.1, and 3.4 M seeds ha$^{-1}$) on winter wheat grain yield and yield components. The seeding rate was nested within row spacing in nested-factorial design. A nested-factorial treatment design was used with testing at several locations in Nebraska across two years. Variety had a substantial effect on winter wheat grain yield ($p < 0.05$). Variety also had a substantial interaction effect with planting date and row spacing 50% of the time ($p \leq 0.01$). At Hemingford, for example, Wesley planted at 19 cm had 5.9% more yield when compared to Robidoux planted at 19 cm (5.5 Mg ha$^{-1}$). Similarly, biomass was influenced by variety across sites ($p < 0.01$), but a substantial interaction effect also occurred between planting date and variety at two of the three sites. Narrow row spacing (19 cm) led to significantly more tillers (6.9 M ha$^{-1}$) when planted with Goodstreak at two of the sites. While planting date by itself did not affect any of the responses evaluated, this research highlights the importance of comprehensive and holistic approaches to wheat production in the High Plains.

**Keywords:** planting date; winter wheat varieties; row spacing; seed rate; wheat yield and biomass

## 1. Introduction

In recent decades, much of the improvement in wheat production has largely been driven by nutrient management and advances in crop genetics [1–4]. In addition, other agronomic practices have long been investigated as a means of enhancing the productivity of food, forage, and energy crops [5–8]. This is expected to continue, and as new cultivars are introduced, it becomes paramount that refinement of agronomic practices is regularly conducted. In particular, dryland cropping systems may require sound agronomic practices to make the best use of limited precipitation [9]. This could help to bridge the yield gap between what is realized, and the yield potential or maximum yield associated with a particular crop cultivar. One aspect of agronomy that can be further explored is the seeding rate for different available genotypes.

Previous research has shown that modern winter wheat cultivars (*Triticum aestivum* L.) may do well under a high seeding rate [10]. As a result, an optimum plant density can lead to efficient use of resources resulting in improved crop productivity [11]. One factor that influences plant density is the tillering potential of a given cultivar [12]. Lower plant density is likely to be used for cultivars with high tillering potential [13]. Conversely,

a density with more plants may be used for genotypes with low tillering potential [12]. However, caution must be taken as plant densities beyond the optimum level increase the potential of lodging resulting from competition for resources resulting in loss of grain yield combined with extra seed costs [14]. Plant density and cultivars also influence yield components such as the number of spikes, spike length, kernels per spike, and 1000-kernel weight [14]. The implication is that cultivars new to a given region need to be evaluated comprehensively to allow producers to make an informed decision regarding plant density.

Sowing date is another agronomic option that producers and researchers investigate to arrive at an appropriate date or length of time over which a cultivar can be successfully grown without compromising on the yield components and grain yield. Early season sowing may optimize precipitation use and nutrients released from the mineralization of soil organic matter [15]. Evidence in Australia suggests that with the right genotypes, it is possible to achieve greater yield with early sowing than the normal optimum planting dates [16]. This is further supported by projections in the UK suggesting that there is an increased likelihood of yield loss for the late sowing date driven by uncertainty in precipitation and temperature [17]. In contrast, Dai et al. [18] showed that delayed sowing improves lodging resistance without compromising on grain yield, kernel weight, and number relative to the early sowing date. These kinds of contradictory conclusions make it necessary for us to ask questions about whether a genotype will be a good fit for known optimum planting dates or not. As a result, re-evaluation of the optimum planting date may be necessary to coincide with a period with sufficient soil moisture to support germination, emergence, and early season growth [19].

Numerous winter wheat genotypes have been released in Nebraska over several decades [20]. These cultivars may differ in straw strength, maturity period, height, coleoptile length, and winter hardiness. For example, Wesley has excellent straw strength and a short plant height [20]. In contrast, Goodstreak and Robidoux both have a straw strength rating of good and with tall and medium plant heights, respectively. These differences among cultivars could dictate the seed rate, spacing, and/ or planting date optimums.

Although agronomic practices of several genotypes of wheat were investigated before over several decades [21–23], there is a potential that climate change may affect some of these agronomic practices to warrant making new recommendations for such practices. One field experiment showed that climate change may make late sowing dates at greater seeding rates reduce the rate of yield decline for winter wheat [24]. The same pattern was also identified by Ding et al. [19]. They suggested that with a changing climate, delaying planting by 10 to 25 days depending on whether it is a wet, medium, or dry year could prove pivotal in sustaining a higher grain yield. A simulated study in the U.S. Great Plains also suggested an improvement in yield and yield components with late planting dates for different cultivars [25]. It is on this basis that planting date, seed rate, crop spacing, and varieties need to be evaluated to make decisions that match with the current evidence. Further, the need to build on the body of knowledge currently represented by few studies that evaluated three or more agronomic practices simultaneously [6,21,22,26] justifies the need for more research. We hypothesize that the four agronomic factors will through a four-, three- or two-way interaction have a significant influence on yield components and the final grain yield.

Therefore, this experiment investigated the influence of four agronomic practices (planting date, seed rate, variety, and spacing) on grain yield and yield components of winter wheat.

## 2. Materials and Methods

The experiments were conducted in Sidney (High Plains Agricultural Laboratory; 41°13′54.5″ N 103°00′48.6″ W), McCook (40°09′31.8″ N 100°45′01.8″ W), and Hemingford (42°14′51.8″ N 103°01′11.1″ W) located in Nebraska (NE). The experiments were set up in 2017 and 2018. The soil at Sidney is classified as Kuma loam (Fine-silty, mixed, superactive, mesic Pachic Argiustolls) with 0–1% slopes. At McCook, the soil is classified as Holdrege

and Keith silt loams (Fine-silty, mixed, superactive, mesic Typic Argiustolls) having 1–3% slopes. The soil classification at Hemingford is Alliance loam (Fine-silty, mixed, superactive, mesic Aridic Argiustolls) with 0–1% slopes.

A nested-factorial treatment design [27] was used in this study. Planting dates, varieties, and row spacing were arranged in a factorial while seed rates were nested within the levels of row spacing. Planting dates included early, optimum (on-time), and late; varieties were Goodstreak, Robidoux, and Wesley; and crop row spacing were 19 and 25 cm. Seed rates were 1.8, 2.1, 2.3, 2.4, 2.6, 2.8, 3.1, and 3.4 M seeds ha$^{-1}$. Since seed rate was nested within the levels of row spacing, the seed rates at the spacing of 19 cm were 2.4, 2.8, 3.1, and 3.4 M seeds ha$^{-1}$. At 25 cm row spacing, the seed rates were 1.8, 2.1, 2.3, and 2.6 M seeds ha$^{-1}$. Each treatment combination was replicated five times. For each site-year, the planting date had no true replication, but the two years of study were used for partitioning its sum of squares. Each experimental unit measured 1.5 m $\times$ 9.1 m with an alley of 1 m between adjacent plots.

The planting dates for winter wheat at the three locations in 2017 and 2018 are shown in Table 1. Planting dates varied by site with a date considered early at one location being an optimum planting date at another location, particularly at McCook where planting tended to be delayed coinciding with optimum planting conditions. This meant that early and late planting are also delayed since they are spread around the optimum planting time. This made it possible for some early, optimum, and late planting dates to overlap, despite being the appropriate timings for each individual location.

**Table 1.** Planting dates in 2017 and 2018 for winter wheat planted at Sidney, Hemingford, and McCook.

| Site | Planting Date | | |
|---|---|---|---|
| | **Early** | **Optimum** | **Late** |
| Sidney | 29 August 2018 | 10 September 2018 | 24 September 2018 |
| | 25 August 2017 | 12 September 2017 | 3 October 2017 |
| Hemingford | 30 August 2018 | 11 September 2018 | 4 October 2018 |
| | 31 August 2017 | 12 September 2017 | 4 October 2017 |
| McCook | 5 September 2018 | 21 September 2018 | 23 October 2018 |
| | 18 September 2017 | 2 October 2017 | 18 October 2017 |

All fields were under no-till production and planted with a no-till drill that has single disk John Deere openers. Nitrogen (N) fertilizer was streamed pre-plant as urea ammonium nitrate (32-0-0) based on soil recommendations for each field belonging to each of the participating growers. On average, growers applied 62 kg N ha$^{-1}$ preplant followed by 11 kg N ha$^{-1}$ as top-dressed urea ammonium nitrate (32-0-0) in the spring during spring weed control at tillering/spring greenup. No phosphorus (P) and potassium (K) fertilizers were applied as the test of soil supply of P (34 mg kg$^{-1}$) and K (132 mg kg$^{-1}$) was considered to be at a sufficient level for winter wheat production. Pre-plant herbicides included burndown with glyphosate (N-(phosphonomethyl)glycine) at 2.3 L ha$^{-1}$ with ammonium sulfate (AMS) and 2,4-D LV-6 (2,4-Dichlorophenoxy acetic acid, 2-ethylhexyl ester) at 0.6 L ha$^{-1}$ applied two weeks before the first planting date. The second herbicide application in the spring was achieved with 2,4-D LV-6 at a rate of 0.6 L ha$^{-1}$ applied during the spring top-dressing of N. Spot weeding to clean alleys was occasionally done with a hand application of 2,4-D or manual hoeing. Crops were produced under a dryland production system without supplemental irrigation.

Tillers were counted at maturity (prior to harvest) from a 30.5 cm row length at two different locations within a plot. From the same samples used for counting tillers, biomass weight and grain yield were measured. From the resulting grains, a 1000-kernel weight for wheat was determined from 1000 randomly selected seeds from each experimental unit. The moisture content of the grain yield was adjusted to 125 g kg$^{-1}$.

Statistical analysis was achieved using PROC GLIMMIX in SAS 9.4 [28]. Since the treatment design was a nested factorial, but the row spacing treatment necessitated the use of strips for the row spacing by seed rate effects, a split plot topographical design was considered in the analysis. Whole plot effect was the planting date, split plot effect included the row spacing within planting date. Split-split plot and nested effect was the seed rate within row spacing and the combination of variety and nested seed rate term.

A linear mixed model was employed using the GLIMMIX Procedure of SAS 9.4 where year and replication within year were treated as random effects. Mean square errors and F-statistics were calculated using the appropriate error terms and degrees of freedom that accounted for both the treatment structure and topographical design aspects. Restricted maximum likelihood (REML) and Gaussian assumptions were used to estimate the variance components of the model. Least squares means (LS-means) were computed using LSMeans statement with a standard error option provided to show the precision of the estimate. Visualization was accomplished in R [29]. Tidyverse, a collection of several R packages, was deployed in the visualization [30].

## 3. Results and Discussion

### 3.1. Grain Yield

At Hemingford, the interaction between planting date, variety, and seed rate nested within row spacing had no significant effect on winter wheat grain yield ($p > 0.05$, Table 2). Apart from planting date × variety, row spacing × variety, and planting date × seed rate nested within row spacing interactions ($p \leq 0.01$), there were no significant effects of two-way, and three-way interactions ($p > 0.05$) on the grain yield. As main effects, variety and row spacing influenced grain yield ($p \leq 0.01$). The interaction between variety and row spacing (Figure 1) meant that the choice of a variety was conditioned on the row spacing used. For instance, Robidoux had 5.4% substantially more grain yield with a row spacing of 19 cm than Goodstreak (5.2 Mg ha$^{-1}$) planted at the same spacing. However, at the spacing of 25 cm, a grower could choose to plant any of the two varieties as there was no significant yield difference between them. Additionally, even though all the varieties yielded more at a spacing of 19 cm, Wesley planted at 25 cm generated a yield (5.5 mg ha$^{-1}$) that significantly exceeded that of Goodstreak planted at a spacing of 19 cm by 5.6%. This illustrates the dependence of variety on row spacing, that is, if planting was to be done at 25 cm, then the choice of a cultivar was likely to be Wesley. Planting cultivars at narrow row spacing led to more yield than wide spacing possibly because close spacing improves the utilization of soil nutrients [31]. This may be because narrow row spacing leads to more spikes per unit area and together with the correct choice of a variety may lead to an improvement in grain yield [32]. Further, the yield tends to be higher for narrowly spaced varieties because of an increase in the number of tillers per unit area [33]. Planting date × variety interaction showed that grain yield was greatest when Wesley was planted at an optimum date (Figure 2a). This yield (6.0 Mg ha$^{-1}$) was significantly greater than the yield of Wesley planted before the optimum planting date (early) and all the cultivars planted during the late planting dates by at least 16.1%.

**Table 2.** Analysis of variance showing the effect of planting date, row spacing, seeding rate, variety, and their interactions on winter wheat grain yield.

| Effect | Num df | Hemingford | McCook | Sidney |
|---|---|---|---|---|
| | | | *p*-values | |
| PD [1] | 2 | NS | NS | NS |
| RS [2] | 1 | 0.0099 | NS | NS |
| SR [3] (RS) | 3 | NS | NS | NS |
| V [4] | 2 | <0.00001 | <0.0001 | 0.02 |
| PD × RS | 2 | NS | NS | NS |
| PD × SR | 6 | 0.016 | NS | NS |
| PD × V | 4 | <0.0001 | <0.0001 | NS |
| RS × V | 2 | 0.0135 | NS | NS |
| SR × V | 6 | NS | NS | NS |
| PD × RS × V | 4 | NS | NS | NS |
| PD × SR(RS) × V | 36 | NS | NS | NS |

[1] PD, planting date. [2] Row spacing. [3] Seed rate. It is nested within row spacing, meaning that every interaction term involving seed rate was nested within row spacing since seed rates were different for each spacing. [4] Variety. NS, not significant at $p > 0.05$. The mean square error of yield was analyzed from data recorded in kg ha$^{-1}$.

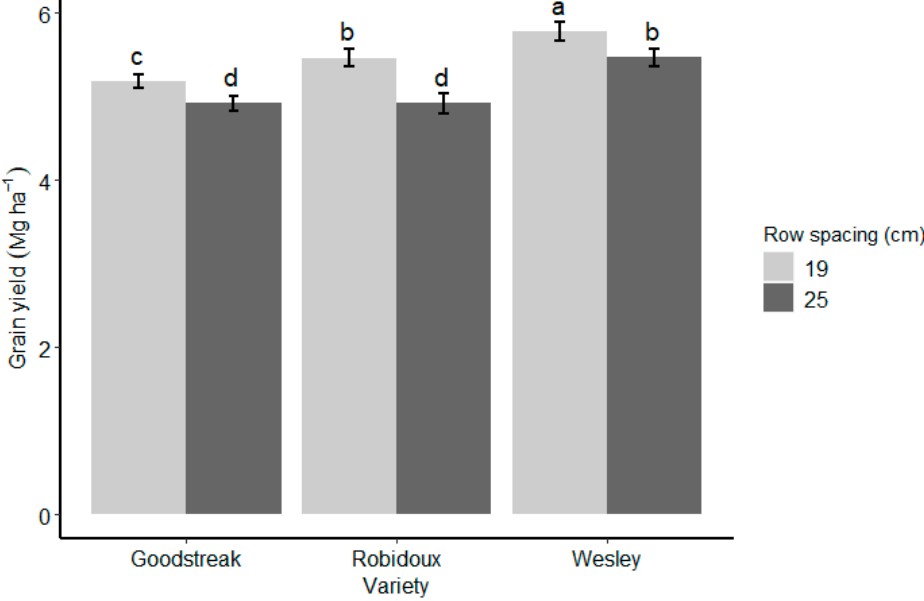

**Figure 1.** Winter wheat grain yield as affected by the interaction between row spacing and variety at Hemingford. The error bars represent the standard error of the mean. Different letters on top of column bars indicate a significant mean difference at $p < 0.05$.

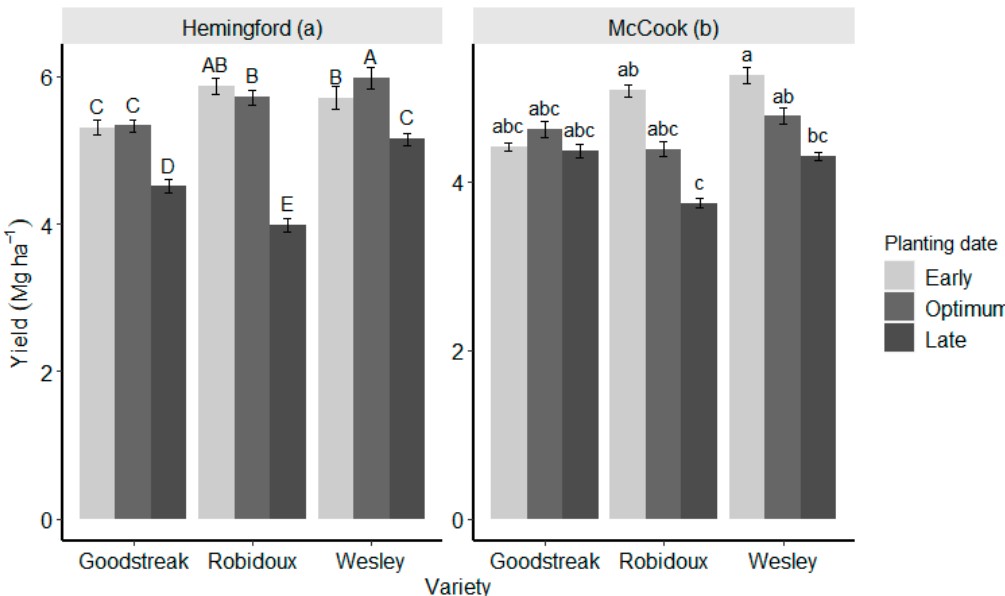

**Figure 2.** Winter wheat grain yield as affected by the interaction between planting date and variety at Hemingford and McCook experimental sites. The error bars represent the standard error of the mean. Different letters on top of column bars of each panel indicate a significant mean difference at $p < 0.05$.

Like results at Hemingford, planting date $\times$ variety $\times$ seeding rate nested within row spacing did not influence the final grain yield at McCook experimental site ($p > 0.05$, Table 2). Planting date and variety interacted to significantly affect the final winter wheat grain yield ($p < 0.01$). The rest of the interaction terms had no major effect on grain yield. For planting date $\times$ variety interaction, the greatest grain yield (5.3 Mg ha$^{-1}$) was obtained with early planting date and Wesley (Figure 2). This grain yield was similar for all the cultivars planted early, on-time or late except for late-planted Wesley and Robidoux whose average yield was 58.3% lower than that of early planted Wesley. Row spacing, as the main effect, did not substantially affect the final grain yield ($p > 0.05$). This meant that both spacings of 19 and 25 cm resulted in a similar grain yield.

The grain yield obtained from Sidney was not substantially affected by the interaction between the different variables considered in this study ($p > 0.05$, Table 2). Among the main effects, it was the variety that affected the grain yield for winter wheat ($p < 0.05$). This result indicated that Goodstreak and Robidoux had similar grain yields averaging 2.9 Mg ha$^{-1}$ (Figure 3). At this site, Wesley had the lowest grain yield which was statistically different from the other two cultivars by at least 4.4%.

Although not in all cases, grain yield appeared not to vary widely among cultivars planted at optimum and early planting dates. However, the choice of a variety becomes important when planting late. Wesley and Goodstreak are apparently better choices when producers contemplate planting late as they yielded greater than Robidoux at Hemingford. Late planting possibly did not allow the crop to accumulate enough growth before the winter dormancy period [34]. Additionally, late planting dates possibly reduced grain yields because of the reduction in kernel weight and the number of kernels per unit area [35,36]. However, as shown in Figure 2a, some varieties will tolerate delayed planting more than others. In this case, Wesley yielded more than Robidoux when both were planted late. Late planting possibly subjects the crop to low temperature during the vegetative growth stage and high temperature during the critical grain filling time [37]. Optimum planting dates increase the possibility of accumulating enough growing degree days before vernalization to promote the transition from the vegetative to reproductive phase [34,38]. This may explain why grain yield was more for varieties planted on time. The height of varieties may also influence the final grain yield [4]. Wesley might have performed better

than other cultivars as it accumulates less biomass due to its short height and directed more resources towards kernels per head.

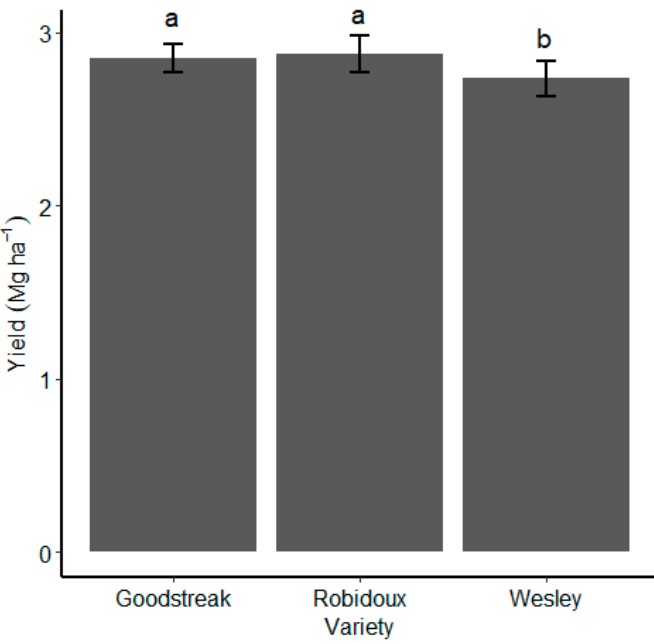

**Figure 3.** Grain yield at Sidney experimental site as affected by variety grown in 2018 and 2019. The error bars represent the standard error of the mean. Different letters on top of column bars indicate a significant mean difference at $p < 0.05$.

*3.2. Biomass*

The interaction between planting date and crop variety significantly affected the quantity of biomass produced by wheat grown at Hemingford ($p < 0.05$; Table 3). Row spacing which had no significant interaction effect with other variables had a substantial effect on wheat biomass when considered as the main effect ($p \leq 0.01$). Apart from the significant interrelationship above, the other interaction terms did not result in biomass differences among treatments ($p > 0.05$). Substantially more biomass (16.2 Mg ha$^{-1}$) was produced by planting winter wheat at a row spacing of 19 cm than at 25 cm (Figure 4). Relative to 19 cm row spacing, the biomass was 19% lower for wheat planted at 25 cm row spacing. A similar result was observed by May et al. [39] who found biomass to be lower at wider spacing. This might be because wider spacing may be less efficient at suppressing weeds when compared to narrow row spacing [40]. This is because wide row spacing provides less ground cover, particularly during early growth, allowing weeds to grow as well as increased evaporation [41]. Alternatively, narrow row spacing led to more efficient use of soil nutrients that translated into more plant biomass [32]. Planting date × variety interaction revealed that the greatest biomass (16.6 Mg ha$^{-1}$) was attained with Robidoux planted at optimum planting dates (Figure 5a). This biomass yield was substantially higher than that of other varieties planted early, on time or late. However, late planting appeared to favor Goodstreak since it had 10% more biomass than Wesley. This drastic difference in biomass was not observed between late-planted Robidoux and Goodstreak.

**Table 3.** Means of main effects, showing significant differences where they exist using letters. Different letters indicate a significant mean difference at $p < 0.05$. For those groups of means with no designation, the levels of the main effects performed similarly across the experiment.

| Main Effect Level | | Hemingford | McCook Mean $\pm$ SE | Sidney |
|---|---|---|---|---|
| **Planting Date** | | | | |
| Early | | 5631 $\pm$ 753 | 4917 $\pm$ 330 | 3461 $\pm$ 1126 |
| On-Time | | 5679 $\pm$ 753 | 4606 $\pm$ 330 | 3008 $\pm$ 1126 |
| Late | | 4548 $\pm$ 753 | 4141 $\pm$ 330 | 1999 $\pm$ 1126 |
| **Row Spacing** | | | | |
| 19 cm | | 5469 $\pm$ 436 [a] | 4781 $\pm$ 214 | 2930 $\pm$ 659 |
| 25 cm | | 5103 $\pm$ 436 [b] | 5328 $\pm$ 214 | 2715 $\pm$ 659 |
| **Seed Rate (Row Spacing)** | | | | |
| 19 cm Row Spacing | 2.4 seeds ha$^{-1}$ | 5354 $\pm$ 439 [bc] | 4584 $\pm$ 219 [bc] | 2868 $\pm$ 661 |
| | 2.8 seeds ha$^{-1}$ | 5431 $\pm$ 439 [ab] | 4713 $\pm$ 219 [b] | 2927 $\pm$ 661 |
| | 3.1 seeds ha$^{-1}$ | 5505 $\pm$ 439 [ab] | 4884 $\pm$ 219 [a] | 2972 $\pm$ 661 |
| | 3.4 seeds ha$^{-1}$ | 5584 $\pm$ 439 [a] | 4940 $\pm$ 219 [a] | 2953 $\pm$ 661 |
| 25 cm Row Spacing | 1.8 seeds ha$^{-1}$ | 5016 $\pm$ 439 [d] | 4220 $\pm$ 220 [c] | 2641 $\pm$ 661 |
| | 2.1 seeds ha$^{-1}$ | 5072 $\pm$ 439 [d] | 4267 $\pm$ 219 [c] | 2763 $\pm$ 661 |
| | 2.3 seeds ha$^{-1}$ | 5146 $\pm$ 439 [d] | 4368 $\pm$ 219 [bc] | 2719 $\pm$ 661 |
| | 2.6 seeds ha$^{-1}$ | 5179 $\pm$ 439 [cd] | 4458 $\pm$ 219 [bc] | 2736 $\pm$ 661 |
| **Variety** | | | | |
| Goodstreak | | 5052 $\pm$ 436 [c] | 4468 $\pm$ 197 [b] | 2867 $\pm$ 654 [a] |
| Robidoux | | 5190 $\pm$ 436 [b] | 4410 $\pm$ 197 [b] | 2867 $\pm$ 654 [a] |
| Wesley | | 5616 $\pm$ 436 [a] | 4786 $\pm$ 197 [a] | 2734 $\pm$ 654 [b] |

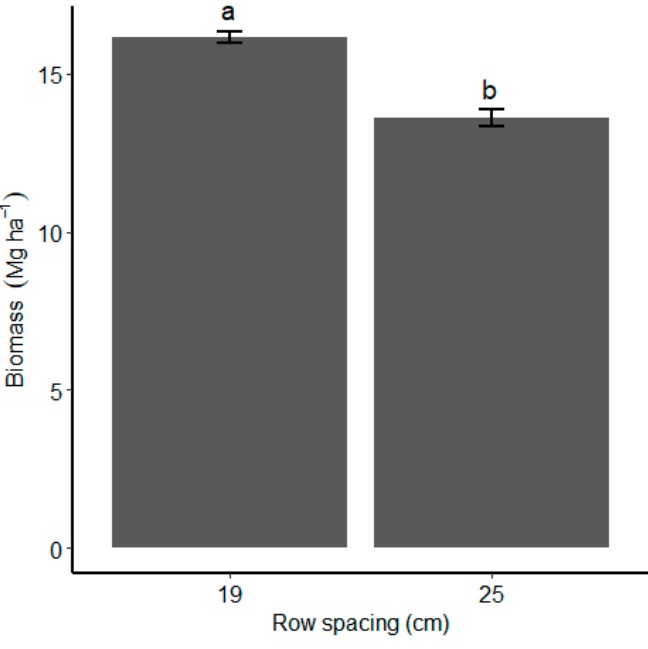

**Figure 4.** Effect of row spacing on the quantity of biomass produced at Hemingford. The error bar represents the standard error of the mean. Different letters on top of column bars indicate a significant mean difference at $p < 0.05$.

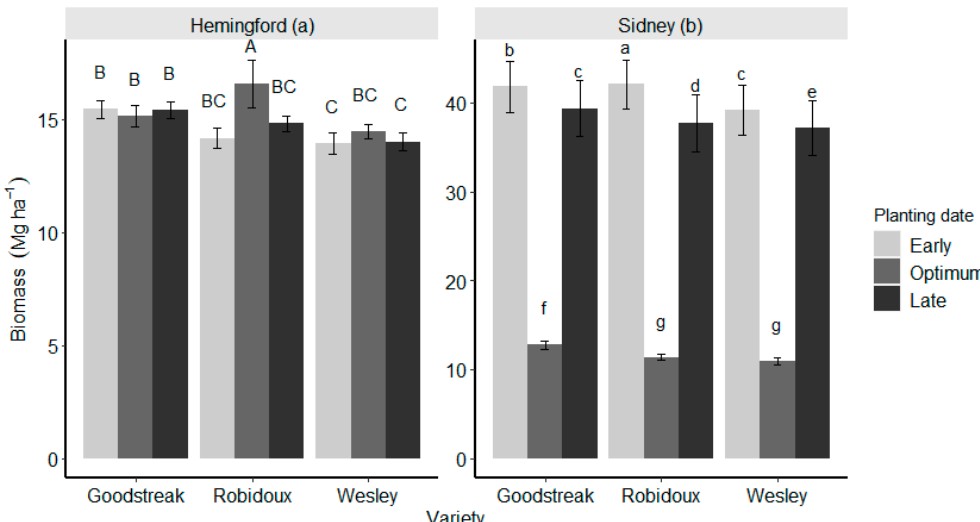

**Figure 5.** Influence of planting date × variety interaction on biomass at Hemingford and Sidney experimental sites. The error bars represent the standard error of the mean. Different letters on top of column bars of each panel indicate a significant mean difference at *p* < 0.05.

With the exception of variety, none of the interactions nor main effects resulted in differences of winter wheat biomass at McCook (*p* < 0.05; Table 4). For varietal differences, the biomass of 12.5 Mg ha$^{-1}$ produced by Goodstreak exceeded the quantity produced by Wesley and Robidoux by an average of 17.9% (Figure 6). The tall height of Goodstreak may be contributing to more biomass accumulated by this variety when compared to Wesley and Robidoux which are short and medium in height, respectively [20]. In addition, Goodstreak is often a mixed purpose wheat that can be used for grain or for hay production. Since the effect of row spacing was not significant, the significant interaction for seed rate nested within row spacing suggests that seed rates at each level of row spacing drove the significant differences in wheat biomass.

Biomass at Sidney experimental site was affected by planting date × variety and row spacing × variety interactions (*p* < 0.05; Table 3). These were the only interaction terms that were significant with the rest of the interactions and main effects of planting date, row spacing, and seed rate nested within row spacing having no significant effect on biomass yield (*p* > 0.05). Results from planting date × variety interaction revealed early planting date and Robidoux as the treatment combination with the greatest biomass of 42.1 Mg ha$^{-1}$ (Figure 5b). All the cultivars performed poorly with optimum planting dates with an average biomass yield of 11.8 Mg ha$^{-1}$. The reason for this phenomenon is unclear as this was expected to mainly affect the late-planted varieties because of a reduction in spikes per unit area [42]. In terms of variety × row spacing, Goodstreak planted at 19 cm had a biomass of 36.6 Mg ha$^{-1}$, a biomass yield that significantly exceeded the biomass produced by Robidoux and 19 cm row spacing treatment combination by 4.3% (Figure 7). Similarly, Goodstreak and Robidoux planted at narrow row spacing (19 cm) had on average 34.8% more biomass than all the varieties planted at a wide row spacing (25 cm). This study showed that a producer wishing to produce more biomass may choose Goodstreak planted at 19 cm because of statistical difference in biomass but at 25 cm, the choice could be either Goodstreak or Robidoux since they had similar biomass.

**Table 4.** Analysis of variance showing the effect of planting date, variety, row spacing, and seed rate nested within row spacing and interaction among them on winter wheat biomass, 1000-kernel weight, and tillering. Location names are abbreviated HEM (Hemingford), MC (McCook), and SID (Sidney).

| Effect | Num df | Biomass | | | 1000-Kernel Weight | | | Tillers | | |
|---|---|---|---|---|---|---|---|---|---|---|
| | | HEM | MC | SID | HEM | MC | SID | HEM | MC | SID |
| | | *p*-Values and Significance Levels | | | | | | | | |
| PD [1] | 2 | NS | NS | NS | NS | NS | NS | NS | NS | NS |
| RS [2] | 1 | 0.0039 | NS | NS | NS | NS | NS | NS | NS | 0.03 |
| SR [3] (RS) | 3 | NS | 0.0436 | NS | NS | NS | NS | NS | NS | NS |
| V [4] | 2 | 0.003 | 0.0029 | <0.0001 | NS | <0.0001 | NS | 0.0003 | <0.0001 | <0.0001 |
| PD × RS | 2 | NS | NS | NS | NS | NS | NS | NS | NS | NS |
| PD × SR(RS) | 6 | NS | NS | NS | NS | NS | NS | NS | NS | NS |
| PD × V | 4 | 0.0417 | NS | <0.0001 | NS | 0.0263 | NS | NS | NS | 0.0055 |
| RS × V | 2 | NS | NS | 0.0291 | NS | NS | NS | NS | NS | 0.0034 |
| V × SR(RS) | 6 | NS | NS | NS | NS | NS | NS | NS | NS | NS |
| PD × RS × V | 4 | NS | NS | NS | NS | NS | NS | NS | NS | NS |
| PD × V × SR(RS) | 36 | NS | NS | NS | NS | NS | NS | NS | NS | NS |

[1] Planting date. [2] Row spacing. [3] Seeding rate. It is nested within row spacing, meaning that every interaction term involving seed rate was nested within row spacing since seed rates were different for each spacing. [4] Variety. NS, not significant at $p > 0.05$. Mean square error of biomass was analyzed in kg ha$^{-1}$ while a 1000-kernel weight was treated as mg 1000 seeds$^{-1}$.

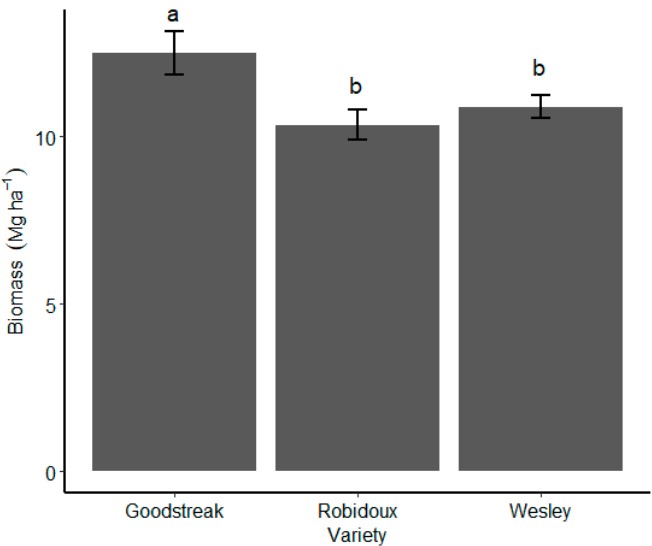

**Figure 6.** The biomass of winter wheat as affected by varieties planted at McCook experimental site. The error bars represent the standard error of the mean. Different letters on top of column bars indicate a significant mean difference at $p < 0.05$.

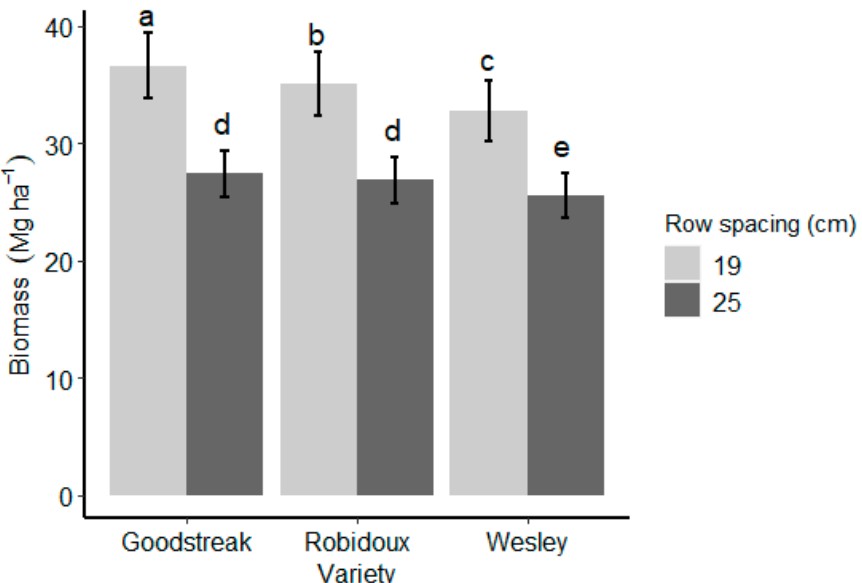

**Figure 7.** The effect of row spacing × variety interaction on the biomass of winter wheat at Sidney. The error bars represent the standard error of the mean. Different letters on top of column bars indicate a significant mean difference at $p < 0.05$.

### *3.3. 1000-Kernel Weight*

A 1000-kernel weight was not influenced by both the main effect and the interactions among several variables considered in this study at Hemingford ($p > 0.05$; Table 4). This showed that a 1000-kernel weight at this site did not depend on planting date, row spacing, variety, seed rate, and/ or the interaction among them at this environment.

Variety and the interrelationship between planting date and variety affected the 1000-kernel weight at McCook experimental site ($p < 0.05$; Table 4). There was no such effect of other main and interaction effects on the 1000-kernel weight at the study site ($p > 0.05$). Wesley planted at an optimum date resulted in the highest weight of 42.1 g per 1000 seeds (Figure 8). This result suggests that planting Wesley at the optimum time will lead to more weight of 1000 seeds and yet if the choice was to plant Robidoux, then early planting time may be more appropriate. Planting at an optimum time particularly for Wesley could lead to an adequate but not excessive number of tillers resulting in more grain weight [43]. Shahzad et al. [44] reported a consistent decrease in a 1000 kernel weight for each delay in planting date and that varieties responded differently. As varieties may be bred to improve a particular crop trait, a 1000 kernel weight may vary among varieties potentially making them respond differently to planting dates. Delayed planting was largely responsible for the lower 1000-kernel weight associated with most varieties [35]. As grain starch content plays an important role in the weight attained by the grain, delayed planting likely reduced its rate of accumulation, and the magnitude varies for each variety [45]. The reduction may be more significant in a warm and dry winter and spring [26]. Therefore, the selection of a variety also needs to consider the time of planting as both interact to affect a 1000-kernel weight.

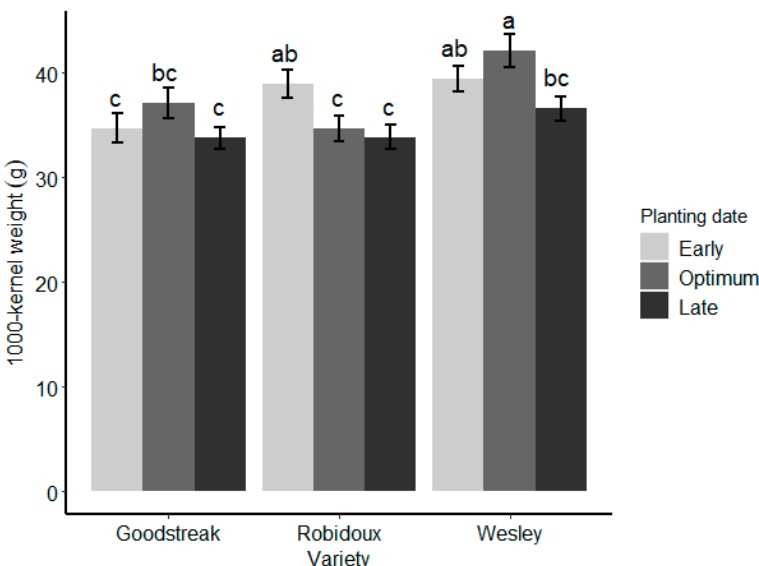

**Figure 8.** Effect of the interaction between planting date and variety on a 1000-kernel weight at McCook. The error bar represents the standard error of the mean. Different letters on top of column bars indicate a significant mean difference at $p < 0.05$.

At Sidney, none of the factors considered influenced a 1000-kernel weight either as main effects or the interaction among them ($p > 0.05$; Table 4). This is similar to observations made at Hemingford. Liu et al. [46] reported that planting date did not affect a 1000-kernel weight. Although a 1000-kernel weight was less influenced by the planting date at the two sites, it is important to view the broader context that grain yield may be reduced due to the reduction in the number of spikes and the production of dry matter and nitrogen [46].

### 3.4. Tillers

A significant effect of the winter wheat variety was observed on the number of tillers produced at Hemingford ($p < 0.01$; Table 4). However, the number of tillers was not influenced by planting date, row spacing, seed rate, and the interaction among the variables. Wesley produced on average 10% fewer tillers when compared to Goodstreak and Robidoux which had an average number of tillers that equaled 6.4 M ha$^{-1}$ (Figure 9).

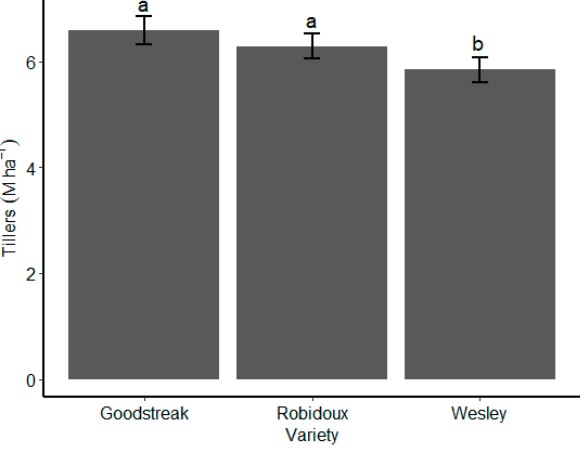

**Figure 9.** The effect of variety on the number of tillers produced at Hemingford. The error bars represent the standard error of the mean. The unit is expressed in million ha$^{-1}$. Different letters on top of column bars indicate a significant mean difference at $p < 0.05$.

At McCook, the main effect of variety led to a significant difference in wheat tillers, while none of the interaction effects had significant impacts on tillering, similar to the

conclusion found in Hemingford for the same trait ($p = 0.05$; Table 4). However, at McCook, Goodstreak had significantly more tillers than either Robidoux or Wesley (data not shown).

In Sidney, row spacing and variety as well as the interactions between planting date and variety, and the individual main effects of row spacing and variety affected the total number of tillers recorded ($p < 0.03$; Table 4). The rest of the treatment combinations and planting date (main effect) only differed in the number of tillers because of a random experimental error ($p > 0.05$). Early planting of Robidoux led to more tillers (7.2 M ha$^{-1}$) and was comparable to that of early planted Goodstreak (Figure 10). Late planting dates for all the varieties led to an average of 50.9% lower number of tillers when compared to early planted Robidoux (Figure 10. However, late-planted Robidoux and Goodstreak still outperformed late-planted Wesley since it had only 4.8 M tillers ha$^{-1}$.

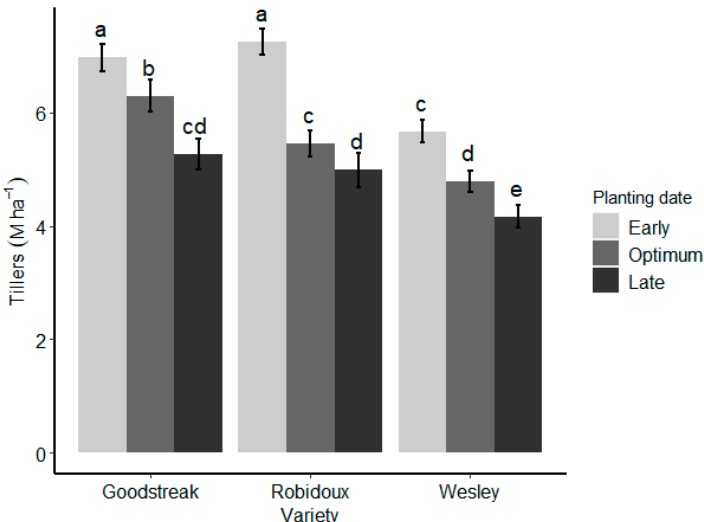

**Figure 10.** Effect of planting × variety interaction on the number of tillers expressed in million ha$^{-1}$ at Sidney. The error bars represent the of the mean. Different letters on top of column bars indicate a significant mean difference at $p < 0.05$.

Row spacing × variety interaction showed a higher number of tillers for closely spaced (19 cm) Goodstreak and Robidoux with an average of 6.9 M tillers ha$^{-1}$ (Figure 10). The two varieties also had an average of 5.3 M tillers ha$^{-1}$ when spaced at 25 cm, a number substantially lower than for a row spacing of 19 cm. Wesley planted at 25 cm resulted in 52.1% lowest number of tillers relative to Goodstreak spaced at 19 cm. Efficient utilization of nutrients, space, and solar radiation may be responsible for why Goodstreak had more tillers under narrow row spacing [47]. Plants attempt to compensate for the wide spacing by producing more tillers but as this study demonstrates, the production of tillers is not adequate to match those produced under narrow spacing. The choice of the right variety and spacing can reduce soil evaporation to ensure more efficient use of the moisture and this may translate into more tillers [41]. In our study, narrow row spacing used with Goodstreak led to a greater number of tillers. Wide spacing is possible to use with all the varieties as they give a related number of tillers.

Early planting of Robidoux and Goodstreak led to more tillers when compared to wheat planted at an optimum time or late in the season. This enhanced tiller formation with these early planted varieties is expected but can also lead to competition and depletion of soil moisture and may not translate to more grain yield [43].

## 4. Conclusions

Overall, evidence from this study suggests that early and optimum planting dates may improve grain yield and that this is often driven by the variety planted. In particular, optimum planting dates may offer some insurance against volatile conditions that may affect early or late-planted crops. Although early planting dates also improved yield, tillers,

1000-kernel weight, and biomass depending on the variety grown, further studies may be necessary to unearth evidence that reinforces its effectiveness in the region. The choice of a variety also depended on the row spacing or planting date and this affected a range of responses including yield, biomass, and tillers. For example, Robidoux attained 5.4% significantly more grain yield at 19 cm row spacing than Goodstreak (5.2 Mg ha$^{-1}$) planted at the same spacing. However, at 25 cm row spacing, any of the two varieties could be grown since they produced identical yields. At two of the three sites, 1000-kernel weight exhibited no response to the main and interaction effects. At the site where 1000-kernel weight was affected by the interaction between variety and planting date, Wesley planted at the optimum time produced the largest 1000-kernel weight (42.1 g per 1000 seeds). However, for early planting, the advantage gained by Wesley planted on-time becomes nonsignificant relative to early planted Robidoux. A disadvantage of planting early, however, is the increased risk of diseases such as wheat streak mosaic virus.

This illustrates the need for considering the interrelationships among these variables in decisions as to which variety to grow at a particular location. Generally, narrow row spacing was more influential on grain yield, biomass, and tillers, suggesting that wider row spacing may have decreased solar radiation utilization efficiency, increased weed competitiveness, or lower soil nutrient and water efficiency. In some instances, the interaction effect was not observed but with significant main effects. This was demonstrated by varietal differences where Goodstreak produced a biomass of 12.5 Mg ha$^{-1}$ and this was 17.9% more biomass than the average quantity produced by Wesley and Robidoux. This could be associated with the tall height of Goodstreak and the short and medium heights of Wesley and Robidoux, respectively. Generally, because of the interrelationship exhibited in this study, the decision about which variety to grow should also consider the planting date, and/or row spacing used at a particular site. Although early planting can have a positive impact on grain yield, early planting can increase the risk of exposure to insect and disease pressure.

**Author Contributions:** Conceptualization, C.F.C. and A.C.E.; methodology, C.F.C. and A.C.E.; formal analysis, A.C.E.; data curation, A.C.E.; writing—original draft preparation, L.A.; writing—review and editing, C.F.C. and A.C.E.; visualization, L.A.; supervision, C.F.C.; project administration, C.F.C.; funding acquisition, C.F.C. All authors have read and agreed to the published version of the manuscript.

**Funding:** This research was partially funded by the Nebraska Wheat Board.

**Data Availability Statement:** Raw data are available upon request to the corresponding author.

**Conflicts of Interest:** The authors declare no conflict of interest.

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
