# Peer review of "Winter Wheat Seeding Decisions for Improved Grain Yield and Yield Components"

_agronomy, doi:10.3390/agronomy12123061_

Round 1
Reviewer 1 Report (Previous Reviewer 2)
The manuscript was clearly improved. However, in my opinion, the authors should improve the conclusions section, in order to be presented more clear and detailed.
Line 22-24: ...nested-factorial design.... or ....split-split plot design....?? Please clarify and correct accordingly.
Author Response
Reviewer 1
The manuscript was clearly improved. However, in my opinion, the authors should improve the conclusions section, in order to be presented more clear and detailed.
Thank you for the input. We pointed out key results as part of the conclusion.
Line 22-24: ...nested-factorial design.... or ....split-split plot design....?? Please clarify and correct accordingly.
Thank you for pointing this out. We changed to a nested-factorial design because the seed rate was nested within row spacing and yet variety, spacing and planting date constituted a complete factorial. This, according to Montgomery, D.C. Design and analysis of experiments. 8 ed. 2013: John wiley & sons., would constitute a nested-factorial design. (http://www.ru.ac.bd/stat/wp-content/uploads/sites/25/2019/03/502_06_Montgomery-Design-and-analysis-of-experiments-2012.pdf).
It is noted in that book on page 616: “Occasionally in a multifactor experiment, some factors are arranged in a factorial layout and other factors are nested”. Because of this, we think that a nested-factorial design is appropriate for our analysis because there is a nested component, that is, seed rate was nested within row spacing and a factorial component, that is, variety, planting date and row spacing.
Reviewer 2 Report (New Reviewer)
An interesting study investigated the effect of planting date, row spacing, variety, and seed rate on winter wheat grain yield and yield components using a nested factorial design.
However, the statistical analysis and the presentation of the results is inappropriate for this type of experiment. Furthermore, the tables are messed up in the text body there are no table legends and there and it's really difficult to understand the results.
Since this is an agronomic paper I suggest that authors present the real means accompanied by standard errors and at the bottom of the tables the p values for the main effects and interactions. I suggest using a 5-way ANOVA with year x planting date x row spacing x variety x seed rate as factors. Replicate should be used as random factors. LME models are good to analyse this type of experiment.
In case of any significant interactions, the interaction means should be presented accompanied by letters to indicate the significance. Authors should also reduce the text (20 pages with numerous tables and 8 figures is too much for a research paper). I am happy to review this manuscript again once it's improved.
Author Response
Reviewer 2
An interesting study investigated the effect of planting date, row spacing, variety, and seed rate on winter wheat grain yield and yield components using a nested factorial design.
However, the statistical analysis and the presentation of the results is inappropriate for this type of experiment.
Thank you for raising this point. You may be right that our statistical analysis is inappropriate, but if you don’t mind, we would refer you to a book written by Montgomery, D.C. Design and analysis of experiments. 8 ed. 2013: John Wiley & Sons (http://www.ru.ac.bd/stat/wp-content/uploads/sites/25/2019/03/502_06_Montgomery-Design-and-analysis-of-experiments-2012.pdf).
It is noted in that book on page 616: “Occasionally in a multifactor experiment, some factors are arranged in a factorial layout and other factors are nested”. Because of this, we think that a nested-factorial design is appropriate for our analysis because there is a nested component, that is, seed rate was nested within row spacing and a factorial component, that is, variety, planting date and row spacing.
Furthermore, the tables are messed up in the text body there are no table legends and there and it's really difficult to understand the results.
Sorry for the mess you found in the manuscript. We discovered that the auto cross-referencing of tables and figures was leading to the insertion of extra tables or figures within the main texts. This has now been removed. We will also submit an additional manuscript in PDF as a supplementary file.
Since this is an agronomic paper I suggest that authors present the real means accompanied by standard errors and at the bottom of the tables the p values for the main effects and interactions.
Thank you for this contribution. If this is not likely to cause a substantial issue with the readers, we would request to maintain how we presented our results with the Figures. Although one may not know the exact value, the Figures have the means and standard errors.
Also, isn’t this likely to make us scatter p-values across many tables? For example, at Hemingford, planting date × variety and row spacing × variety are all significant interaction terms (Table 2). It looks like it would be difficult to include the mean values for these two interactions in the same table together with their p-values. We appreciate it if you allow the current format of the presentation of results to stand.
I suggest using a 5-way ANOVA with year x planting date x row spacing x variety x seed rate as factors. Replicate should be used as random factors. LME models are good to analyse this type of experiment.
In case of any significant interactions, the interaction means should be presented accompanied by letters to indicate the significance. Authors should also reduce the text (20 pages with numerous tables and 8 figures is too much for a research paper). I am happy to review this manuscript again once it's improved.
We appreciate your contribution to making this a sound article. We treated both the year and replication as random effects with the GLIMMIX model which just like LME, assumes normal random effects. These random variables (year and replication) which we missed including a description of, are now reflected under materials and methods.
The 5-way interaction you suggested may be problematic in the sense that the levels of seed rate are not the same for the different levels of row spacing, that is, at a spacing of 19 cm, the seed rates were 2.4, 2.8, 3.1, and 3.4 M seeds ha-1. At 25 cm row spacing, the seed rates were 1.8, 2.1, 2.3, and 2.6 M seeds ha-1. This means we cannot perform the normal complete factorial and again, this explains our choice for a nested-factorial design.
We also understand your concern about having many pages. This is because we also looked at many explanatory (4) and response (4) variables and considering that, in some cases, we have significant interaction terms while others had only significant main effects, we ended up having many pages (18). Above all, other reviewers requested the expansion of some sections, particularly the discussion of results.

Round 2
Reviewer 1 Report (Previous Reviewer 2)
The manuscript has been improved from the original version and, in my opinion, it warrants publication in agronomy.
Author Response
We appreciate your time and support. Thank you!
Reviewer 2 Report (New Reviewer)
Please find comments in attached pdf file

Author Response
Please see attached document. Thank you

This manuscript is a resubmission of an earlier submission. The following is a list of the peer review reports and author responses from that submission.
Round 1
Reviewer 1 Report
Please see attached document for comments and inputs that may help to improve the manuscript.

Reviewer 2 Report
The authors propose a manuscript titled “Winter Wheat Seeding Decisions for Improved Grain Yield and Yield Components”.
I suggest the following changes:
In line 143 title should be changed to Results and Discussion
The discussion section should be further enriched with additional bibliography
What is the use of the different p value estimates? why not use one?
Please provide new ANOVA tables with the mean squares and the significance with ** or * for 99% and 95% significance, respectively
Round 2
Reviewer 2 Report
Dear authors,
a lot of programs provide ANOVA tables with mean squares, eg. SPSS
The authors limited the improvement of the discussion section only with regard to two of the four parameters studied in this paper (grain yield, biomass). I suggest improving the other two paragraphs as well.
Author Response
Thank you for providing valuable feedback for our manuscript. We have addressed each of your comments below.
1) A lot of programs provide ANOVA tables with mean squares, eg. SPSS
Response: We appreciate the suggestion, but respectfully disagree. Mean squares could be included in tables, but would not contribute additional useful information beyond what is already provided in the summary table of p-values and least square means. Instead, we feel that their inclusion would decrease the interpretability of results by making the tables less accessible for readers.
2) The authors limited the improvement of the discussion section only with regard to two of the four parameters studied in this paper (grain yield, biomass). I suggest improving the other two paragraphs as well.
Response: We provided additional information on 1000-kernal weight and tillers.
Thank you